# Galactic Wormhole under Lovelock Gravity

Koushik Chakraborty [1], Farook Rahaman [2,*], Saibal Ray [3], Banashree Sen [4] and Debabrata Deb [5]

1 Department of Physics, Government College of Education, Burdwan 713102, West Bengal, India
2 Department of Mathematics, Jadavpur University, Kolkata 700032, West Bengal, India
3 Centre for Cosmology, Astrophysics and Space Science (CCASS), GLA University, Mathura 281406, Uttar Pradesh, India
4 Department of Applied Mathematics, Maulana Abul Kalam Azad University of Technology, Haringhata 741249, West Bengal, India
5 Department of Physics, The Institute of Mathematical Sciences, Chennai 600113, Tamil Nadu, India
* Correspondence: rahaman@associates.iucaa.in

**Abstract:** We explore wormhole geometry in spiral galaxies under the third order Lovelock gravity. Using the cubic spline interpolation technique, we find the rotational velocity of test particles in the halo region of our spiral galaxy from observed values of radial distances and rotational velocities. Taking this value of the rotational velocity, we are able to show that it is possible to present a mathematical model regarding viable existence of wormholes in the galactic halo region of the Milky Way under the Lovelock gravity. A very important result that we obtain from the present investigation is that galactic wormhole in the halo region can exist with normal matter as well as exotic matter.

**Keywords:** galatic halo; wormhole; Lovelock gravity

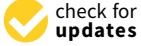

## 1. Introduction

The seed of a hypothetical geometric object connecting two asymptotically flat space-time regions was featured in the works of Weyl [1], and Einstein and Rosen [2]. Much later, it was termed as Wormhole (WH) [3] and the existence of such Einstein–Rosen bridge in the form of traversable wormhole was first ever firmly proposed by Morris and Thorne [4]. They assumed a stress–energy tensor for these wormholes that violates the standard energy conditions. Since matter that violates null energy condition is exotic matter, traversable wormhole solutions need to assume the existence of exotic matter. An important condition for wormholes is that they have no event horizon. There exists extensive literature exploring aspects of traversable wormholes (See [5,6] for review).

Studying WH in modified and higher dimensional gravity theories is also an active field of research since long time. Shang and Xu [7] confirmed the existence of WH solution in general in Lovelock gravity. Bhawal and Kar [8] found the static WH solution in second order Lovelock gravity, i.e., Einstein–Gauss–Bonnet gravity. Bandypadhyay and Chakraborty [9] constructed spherically symmetric thin shell WH in Einstein–Yang–Mills–Gauss–Bonnet gravity. They showed that for certain choices of parameters involved ordinary matter is sufficient for the formation of thin shell WH. Lobo and Oliviera [10] explored possible WH solutions in the context of $f(R)$ modified theories of gravity. Considering violation of Weak Energy Condition (WEC) they obtained the possible range of values for the coupling parameter $\omega$. Dehghani and Dayyani [11] obtained $n$-diemnsional Lorentzian WH solutions of third order Lovelock gravity. They found that the WH throat radius has a lower limit that depends on the Lovelock coefficients, the dimensionality of the spacetime and the shape function. They also derived the region in which the throat could be constituted from the normal matter. Böhmer et al. [12] explored the possibility of static spherically symmetric traversable WH solutions in modified teleparallel gravity.

On the other hand, Mehdizadeh and Riazi [13] proposed dynamic WH solutions in the framework of Lovelock gravity with compact extra dimensions. Harko et al. [14] investigated the WH solutions in which the matter constituting the wormhole throat satisfied all the energy conditions in the context of modified gravity theories. In particular they explored $f(R)$ gravity, the curvature–matter coupling and the $f(R, \mathcal{L}_m)$ generalization. In their article Mehdizadeh, Zangeneh and Lobo [15] obtained thin shell WH solutions in third order Lovelock gravity using cut and paste technique. They explored different situations depending on choices of second and third order Lovelock gravity coefficients. Zangeneh et al. [16] explored the possibilities for traversable WHs in third order Lovelock gravity with a cosmological constant term in an $n$-dimensional spacetime $\mathcal{M}^4 \times \mathcal{K}^{n-4}$ where $\mathcal{K}^{n-4}$ is a constant curvature space. Mehedizadeh and Lobo [17] studied wormhole geometries in third order Lovelock gravity with specifically chosen redshift function and equation of state. There are plethora of articles exploring various aspects of WH geometry in different higher dimensional and modified gravity theories [18–28].

In the recent past, Rahaman et al. [29] considering the Navarro–Frenk–White (NFW) [30,31] density profile confirmed the possible existence of WH spacetime in the outer regions of galactic halo from general relativistic framework. Similar observation was also reported by Kuhfittig [32]. In another seminal paper, Rahaman et al. [33] used the Universal Rotation Curve (URC) [34] dark matter model in the galactic halo region to obtain analogous results for the central parts of the halo. The authors generalized the result to predict possible existence of wormholes in most of the spiral galaxies. In yet other papers, Rahaman et al. [35,36] taking NFW density profile as well as the URC and wormhole such as line element as input calculated the tangential velocities $v^\phi$ of the test particles in the galactic halo. They reported a satisfactory matching of the theoretical and observational plot in the range $9 \, kpc \le r \le 100 \, kpc$.

In the present paper, we extend the works of Rahaman et al. [29,33,35,36] to construct galactic wormhole model in the context of Lovelock gravity. Under this motivation our plan is as follows: In the Section 2 we discuss the basic principles of Lovelock gravity theories. In Section 3, formulation of the problem is provided, while the results and discussions are presented in Section 4. Finally, in Section 5, we conclude with specific comments on the results of the study and future prospects.

## 2. Brief Outline of Lovelock Gravity Theory

The action in the framework of third-order Lovelock gravity, is given by

$$I = \int d^n x \sqrt{-g} \left( \mathcal{L}_1 + \alpha'_2 \mathcal{L}_2 + \alpha'_3 \mathcal{L}_3 \right) \tag{1}$$

where we assumed $8\pi G_n = 1$, $G_n$ being the $n$-dimensional gravitational constant. Here $\alpha'_2$ and $\alpha'_3$ are the second (Gauss–Bonnet) and third order Lovelock coefficients, $g$ is the determinant of the metric, $\mathcal{L}_1 = R$ is the Einstein–Hilbert Lagrangian, the term $\mathcal{L}_2$ is the Gauss–Bonnet Lagrangian given by

$$\mathcal{L}_2 = R_{ijkl}R^{ijkl} - 4R_{ij}R^{ij} + R^2, \tag{2}$$

and the third order Lovelock Lagrangian $\mathcal{L}_3$ is defined as

$$
\begin{aligned}
\mathcal{L}_3 &= 2R^{ijkl}R_{klmn}R^{mn}{}_{ij} + 8R^{ij}{}_{km}R^{kl}{}_{jn}R^{mn}{}_{il} \\
&\quad + 24R^{ijkl}R_{kljm}R^m{}_i + 3RR^{ijkl}R_{klij} + 24R^{ijkl}R_{ki}R_{lj} \\
&\quad + 16R^{ij}R_{jk}R^k{}_i - 12RR^{ij}R_{ij} + R^3.
\end{aligned}
\tag{3}
$$

In Lovelock theory, for an $n$-dimensional space, only terms with order less than $\left[\frac{(n+1)}{2}\right]$ contribute to the field equations. Here, we used the notation that $\left[\frac{n}{2}\right]$ will give biggest

integer less than $\frac{n}{2}$. Since we are considering third order Lovelock gravity, its effects will be apparent for $n \geq 7$.

Thus, varying the action (1) with respect to the metric we obtain the field equations up to third order as follows:

$$G_{ij}^E + \alpha_2' G_{ij}^{(2)} + \alpha_3' G_{ij}^{(3)} = T_{ij}, \tag{4}$$

where $T_{ij}$ is the energy–momentum tensor, $G_{ij}^E$ is the Einstein tensor whereas $G_{ij}^{(2)}$ and $G_{ij}^{(3)}$ are given by

$$
\begin{aligned}
G_{ij}^{(2)} ={} & 2(-R_{ikln}R^{lnk}{}_j - 2R_{imjk}R^{mk} - 2R_{ik}R^k{}_j + RR_{ij}) \\
& - \frac{1}{2}\mathcal{L}_2 g_{ij}, \\
G_{ij}^{(3)} ={} & -3(4R^{nmkl}R_{klpm}R^p{}_{jni} - 8R^{nm}{}_{pk}R^{kn}{}_{ni}R^p{}_{jml} \\
& + 2R_j{}^{nkl}R_{klpm}R^{pm}{}_{ni} - R^{nmkl}R_{klnm}R_{ji} \\
& + 8R^n{}_{jkm}R^{kl}{}_{ni}R^m{}_l + 8R^k{}_{jnl}R^{nm}{}_{ki}R^l{}_m \\
& + 4R_j{}^{nkl}R_{klim}R^m{}_n - 4R_j{}^{nkl}R_{klnm}R^m{}_i \\
& + 4R^{nmkl}R_{klni}R_{jm} + 2RR_j{}^{lnm}R_{nmli} \\
& + 8R^n{}_{jim}R^m{}_k R^k{}_n - 8R^k{}_{jnm}R^n{}_k R^m_i \\
& - 8R^{nm}{}_{ki}R^k{}_n R_{jm} - 4RR^n{}_{jim}R^m{}_n + 4R^{nm}R_{mn}R_{ji} \\
& - 8R^n{}_j R_{nm}R^m{}_i + 4RR_{jm}R^m{}_i \\
& - R^2 R_{ji}) - \frac{1}{2}\mathcal{L}_3 g_{ij}.
\end{aligned}
$$

## 3. Basic Equations and Their Solutions

The $n$-dimensional traversable wormhole metric is given by

$$ds^2 = -e^{2\phi(r)}dt^2 + \left(1 - \frac{b(r)}{r}\right)^{-1}dr^2 + r^2 d\Omega_{n-2}^2, \tag{5}$$

using units in which $c = G = 1$. Here $\phi(r)$ is the red-shift function which must be everywhere finite to prevent the event horizon, $b(r)$ is the shape function and $d\Omega_{n-2}^2$ is the metric on the surface of a $(n-2)$-sphere. The shape function of the wormhole essentially satisfies the condition $b(r_{th}) = r_{th}$ at $r = r_{th}$ where $r_{th}$ is the throat of the wormhole. This condition is commonly known as the flare-out condition which gives at the throat $b'(r_{th}) < 1$ while $b(r) < r$ near the throat.

The energy–momentum tensor is given by

$$T_\nu^\mu = diag[-\rho(r), p_r(r), p_t(r), p_t(r), \ldots]. \tag{6}$$

where $\rho(r)$ is the energy density, $p_r(r)$ is the radial pressure and $p_t(r)$ gives the transverse pressure.

The Einstein equations for the above mentioned metric are as follows [17]

$$
\begin{aligned}
\rho(r) ={} & \frac{(n-2)}{2r^2}\left[-\mathcal{B}_2(r)\frac{(b-rb')}{r}\right] \\
& + \frac{(n-2)b}{2r^3}\left[(n-3) + (n-5)\frac{\alpha_2 b}{r^3} + (n-7)\frac{\alpha_3 b^2}{r^6}\right],
\end{aligned} \tag{7}
$$

$$p_r(r) = \frac{(n-2)}{r}\left[\mathcal{B}_1(r)\mathcal{B}_2(r)\phi'\right]$$

$$- \frac{(n-2)b}{2r^3}\left[(n-3)+(n-5)\frac{\alpha_2 b}{r^3}+(n-7)\frac{\alpha_3 b^2}{r^6}\right], \tag{8}$$

$$p_t(r) = \mathcal{B}_1(r)\mathcal{B}_2(r)\left[\phi''+\phi'^2+\frac{(b-rb')\phi'}{2r(r-b)}\right]$$

$$-\frac{2\phi'}{r^4}\mathcal{B}_1(r)(b-b'r)\left(\alpha_2+3\alpha_3\frac{b}{r^3}\right)$$

$$+\mathcal{B}_1(r)\mathcal{B}_3(r)\left[(n-3)+(n-5)\frac{2\alpha_2 b}{r^3}+(n-7)\frac{3\alpha_3 b^2}{r^6}\right]$$

$$-\frac{b}{2r^3}(n-3)(n-4)+(n-5)(n-6)\frac{\alpha_2 b^2}{2r^6}$$

$$+(n-7)(n-8)\frac{\alpha_3 b^3}{2r^9}, \tag{9}$$

where $\alpha_2 = (n-3)(n-4)\alpha_2'$, $\alpha_3 = (n-3)(n-4)(n-5)(n-6)\alpha_3'$, $\mathcal{B}_1(r) = \left(1-\frac{b}{r}\right)$, $\mathcal{B}_2(r) = \left(1+\frac{2\alpha_2 b}{r^3}+\frac{3\alpha_3 b^2}{r^6}\right)$ and $\mathcal{B}_3(r) = \left(\frac{\phi'}{r}+\frac{(b-rb')}{2r^2(r-b)}\right)$. In the above equations, the prime denotes the derivative with respect to $r$.

From the stable circular geodesic motion in the equatorial plane, the tangential component of the rotational velocity of the neutral hydrogen clouds in the galactic halo region can be found out from the gravitational potential corresponding to the flat rotation curves in the halo region as [29,33,35,36]

$$(v^t)^2 = r\phi'(r). \tag{10}$$

Logically, it is plausible that if an observer is sitting in the plane, $\theta_3$ = constant, $\theta_4$ = constant, ..., $\theta_n$ = constant, then the observer recognizes all characteristics as a (3 + 1) dimensional picture. In this sense, one can use all the relevant data which are approximately the same for both space-times [37].

Moreover, it is generally believed that interactions other than gravity are confined to $3 + 1$ brane. However, gravity can propagate in all extra dimensions ($n \geq 4$) [38,39]. These "gravity-only" dimensions (GODs) may be compactified [40,41]. In the present study of wormholes, we used observational data of tangential velocities of neutral hydrogen cloud. These velocities are determined by the effective gravitational potential in the galactic halo. It must be influenced by the existence of higher dimensions, $n \geq 4$. Therefore, there must not be any problem in using the aforementioned data in the present study. Similar observational data was being used in many other important studies of higher dimensions [42,43].

We would also like to add here that in the literature [44–47] some works are available where the authors try to find out a higher dimensional connection with microphysics and/or macroastrophysics. Especially, Barnafoldi et al. [46] displayed the detection of electro-magnetic and particle radiation from the direction of $Cygnus X3$, which raise the question of the existence of special long-lived, neutral particles. After investigation of the origin of these particles they concluded that the source object may contain compactified extra dimension and these particles are messengers of this state.

Taking into consideration of all the aforementioned points, we use the data in Table 1 in the context of third order Lovelock gravity. From the best fit with the observational data, we obtain the following expression for $v^t$ as

$$v^t = \sum_{i=0}^{i=5} c_i r^i, \tag{11}$$

where $c_i, i = 0\ldots 5$ are constants having values $c_5 = 0.0000011$, $c_4 = -0.0003$, $c_3 = 0.031$, $c_2 = -1.4$, $c_1 = 28$ and $c_0 = 64$.

**Table 1.** The radial distance from galactic centre $r$ in kpc and velocity $v^t$ in km/s of objects in the galactic halo region with total virial mass $3 \times 10^{12} M_\odot$ [35,36,48,49].

| $R$ (kpc) | $v^t$ (km/s) | $R$ (kpc) | $v^t$ (km/s) | $R$ (kpc) | $v^t$ (km/s) |
|---|---|---|---|---|---|
| 0.1 | 10.053 | 34.1 | 234.623 | 68.1 | 234.293 |
| 1.1 | 74.467 | 35.1 | 234.225 | 69.1 | 234.317 |
| 2.1 | 118.223 | 36.1 | 233.891 | 70.1 | 234.334 |
| 3.1 | 151.113 | 37.1 | 233.390 | 71.1 | 234.345 |
| 4.1 | 176.445 | 38.1 | 233.213 | 72.1 | 234.349 |
| 5.1 | 196.099 | 39.1 | 233.079 | 73.1 | 234.347 |
| 6.1 | 211.331 | 40.1 | 232.982 | 74.1 | 234.339 |
| 7.1 | 223.057 | 41.1 | 232.918 | 75.1 | 234.324 |
| 8.1 | 231.975 | 42.1 | 232.883 | 76.1 | 234.303 |
| 9.1 | 238.634 | 43.1 | 232.873 | 77.1 | 234.303 |
| 10.1 | 243.475 | 44.1 | 232.884 | 78.1 | 234.276 |
| 11.1 | 246.857 | 45.1 | 232.913 | 79.1 | 234.243 |
| 12.1 | 249.072 | 46.1 | 232.957 | 80.1 | 234.205 |
| 13.1 | 250.362 | 47.1 | 233.013 | 81.1 | 234.160 |
| 14.1 | 250.927 | 48.1 | 233.078 | 82.1 | 234.109 |
| 15.1 | 250.930 | 49.1 | 233.151 | 83.1 | 234.054 |
| 16.1 | 250.509 | 50.1 | 233.230 | 84.1 | 233.993 |
| 17.1 | 249.774 | 51.1 | 233.313 | 85.1 | 233.927 |
| 18.1 | 248.817 | 52.1 | 233.397 | 86.1 | 233.856 |
| 19.1 | 247.712 | 53.1 | 233.483 | 87.1 | 233.779 |
| 20.1 | 246.519 | 54.1 | 233.568 | 88.1 | 233.698 |
| 21.1 | 245.285 | 55.1 | 233.652 | 89.1 | 233.612 |
| 22.1 | 244.048 | 56.1 | 233.733 | 90.1 | 233.522 |
| 23.1 | 242.836 | 57.1 | 233.811 | 91.1 | 233.427 |
| 24.1 | 241.671 | 58.1 | 233.733 | 92.1 | 233.328 |
| 25.1 | 240.568 | 59.1 | 233.811 | 93.1 | 233.225 |
| 26.1 | 239.539 | 60.1 | 233.886 | 94.1 | 233.118 |
| 27.1 | 238.589 | 61.1 | 233.956 | 95.1 | 233.007 |
| 28.1 | 237.724 | 62.1 | 234.021 | 96.1 | 232.892 |
| 29.1 | 236.943 | 63.1 | 234.081 | 97.1 | 232.774 |
| 30.1 | 236.245 | 64.1 | 234.136 | 98.1 | 232.652 |
| 31.1 | 235.628 | 65.1 | 234.184 | 99.1 | 232.527 |
| 32.1 | 235.089 | 66.1 | 234.227 | 0 | 0 |
| 33.1 | 234.623 | 67.1 | 234.263 | 0 | 0 |

## 4. Results and Discussion

Here, we shall calculate the expression of redshift function $\phi(r)$ by using the expression for $v^t$ given in (11). With the help of Equations (10) and (11), we obtain

$$\phi'(r) = c_5^2 r^9 + 2c_4 c_5 r^8 + \left(c_4^2 + 2c_3 c_5\right) r^7 + 2(c_3 c_4 + c_2 c_5) r^6$$

$$+\left(c_3^2 + 2c_2 c_4 + 2c_1 c_5\right) r^5 + 2(c_2 c_3 + c_1 c_4 + c_0 c_5) r^4 +$$

$$\left(c_2^2 + 2c_1 c_3 + 2c_0 c_4\right) r^3 + 2(c_1 c_2 + c_0 c_3) r^2$$

$$+ \left(c_1^2 + 2c_0 c_2\right) r + \frac{c_0^2}{r} + 2c_0 c_1, \tag{12}$$

$$\phi(r) = \frac{1}{10} c_5^2 r^{10} + \frac{2}{9} c_4 c_5 r^9 + \frac{1}{8}\left(c_4^2 + 2c_3 c_5\right) r^8$$

$$+\frac{2}{7}(c_3 c_4 + c_2 c_5) r^7 + \frac{1}{6}\left(c_3^2 + 2c_2 c_4 + 2c_1 c_5\right) r^6$$

$$+\frac{2}{5}(c_2 c_3 + c_1 c_4 + c_0 c_5) r^5 + \frac{1}{4}\left(c_2^2 + 2c_1 c_3 + 2c_0 c_4\right) r^4$$

$$+\frac{2}{3}(c_1c_2 + c_0c_3)r^3 + \frac{1}{2}\left(c_1^2 + 2c_0c_2\right)r^2$$

$$+ 2c_0c_1r + c_0^2\log(r), \tag{13}$$

$$\phi''(r) = 9c_5^2r^8 + 16c_4c_5r^7 + 7\left(c_4^2 + 2c_3c_5\right)r^6$$

$$+12(c_3c_4 + c_2c_5)r^5 + 5\left(c_3^2 + 2c_2c_4 + 2c_1c_5\right)r^4$$

$$+8(c_2c_3 + c_1c_4 + c_0c_5)r^3 + 3\left(c_2^2 + 2c_1c_3 + 2c_0c_4\right)r^2$$

$$-\frac{c_0^2}{r^2} + 4(c_1c_2 + c_0c_3)r + c_1^2 + 2c_0c_2. \tag{14}$$

Now, we can substitute Equations (12)–(14) in Equations (7)–(9). We obtain very nonlinear equations of the function $b(r)$. In the next two sections, therefore, we take some specific forms of the shape function and study the nature of the spacetime under consideration. The 4-dimensional present spacetime structure is believed to be the self-compactified form of manifold with multidimensional spacetime. It is observed that cosmic string as well as superstring theories and hence M-theory reproduce higher dimensional general relativity at low energy so that scientists have been argued that theories of unification tend to require extra spatial dimensions to be consistent with the physically viable models [50–55]. Several authors have shown that some features of higher dimensional black holes differ significantly from four dimensional black holes which is due to the fact that higher dimensional analysis provides a much wider avenue to black hole solutions in comparison to 4-dimensional counterparts [56–59]. In the present investigation of wormholes, we considered third order Lovelock gravity. Therefore, we took $n \geq 7$. Cases for $n = 8, 9, 10, 11$ are studied. In these studies, all extra dimensions are taken to be compactified.

*4.1. Case I: Physical Features for the Shape Function $b(r) = r\left(\frac{r_0}{r}\right)^k$*

For general physical reasons, the shape function $b(r)$ must obey the following conditions to ensure a traversability of wormholes [26,60]:

(i)　Throat condition: at the throat (i.e., $r = r_0$), the shape function should satisfy the condition $b(r_0) = r_0$, and for $r > r_0$ one should obtain $1 - \frac{b(r)}{r} > 0$.

(ii)　Flaring out condition: this condition dictates that $b'(r_0) < 1$.

(iii)　Asymptotically flatness condition: for this condition we should have $\frac{b(r)}{r} \to o$ as $r \to \infty$.

Under the above mentioned three conditions, we consider the shape function as follows:

$$b(r) = r\left(\frac{r_0}{r}\right)^k, \tag{15}$$

where $r_0$ is throat radius and $k$ is arbitrary parameter. From the flare out condition of the wormhole, it may be found out that $k > 0$. We assume $k = 2$ in the present case as a representative value. With this choice of the shape function, let us look into the physical features of various quantities of interest:

(1)　Now, the expression of the redshift function calculated from the consideration of the galactic flat rotation curves (See Figure 1) as given in Equation (12) remains finite everywhere. This clearly implies that there would be no event horizon which is an important criterion for the existence of wormholes.

(2)　Plot on the left panel of Figure 2 show that the present shape function satisfies the condition $\frac{b(r)}{r} < 1$ for $r > r_{th}$. This indicates that the flare out condition of the wormhole $b'(r_{th}) < 1$ is satisfied.

(3)　In terms of principal pressures the Null Energy Condition (NEC) is given by following inequations:

$$\rho + p_r \geq 0, \tag{16}$$
$$\rho + p_t \geq 0, \tag{17}$$

whereas Weak Energy Condition (WEC) is given by following inequations:

$$\rho \geq 0, \tag{18}$$
$$\rho + p_r \geq 0, \tag{19}$$
$$\rho + p_t \geq 0. \tag{20}$$

Thorne [4] observed that the traversable wormholes violate the Null Energy Condition (NEC) near the throat. In Figure 3, $\rho + p_t$ is plotted for four possible choices of $\alpha_2$ and $\alpha_3$, i.e., $(\alpha_2 > 0, \alpha_3 > 0)$, $(\alpha_2 > 0, \alpha_3 < 0)$, $(\alpha_2 < 0, \alpha_3 > 0)$ and $(\alpha_2 < 0, \alpha_3 < 0)$. The plots clearly show that $\rho + p_t > 0$ for $0 \leq r \leq 50$ kpc and $n$ does not influence the nature of the plots. Thus, the NEC is satisfied in the galactic halo region. In Figure 4 similar plots are noted for $\rho + p_r$. No violation of the NEC is noted here also. Moreover, since $\rho > 0$ in the galactic halo region, the matter threading the wormhole satisfies the WEC (Figure 5). Studies investigating traversable wormhole in the galactic halo region under general theory of relativity, reported violation of the NEC near the throat indicating the existence of exotic matter [29,32,33,35,36]. In the context of Lovelock gravity; however, there are exceptions. Dehghani and Dayyani [11] reported wormhole solutions in third order Lovelock gravity satisfying the WEC. They further showed that positivity of the density and $p + \rho$ depends on the Lovelock coefficients. Mehdizadeh and Riazi [13] studied wormhole solution under Lovelock gravity supported by normal matter. There are numerous other studies concluding the existence of normal matter near the throat of the wormhole in second or third order Lovelock gravity [17,61].

(4)　Figure 5 shows the variation of density function with distance in the galactic halo region. The plots show very highly dense region near the centre of the galaxy and its value very quickly decreases to a small value with increasing distance. The variation is nearly inverse square in the outer region of the galactic halo. However, the density function remains positive in the galactic halo region. These features of the density profile are in accordance with the predictions by many earlier studies of the galactic halos [62–65].

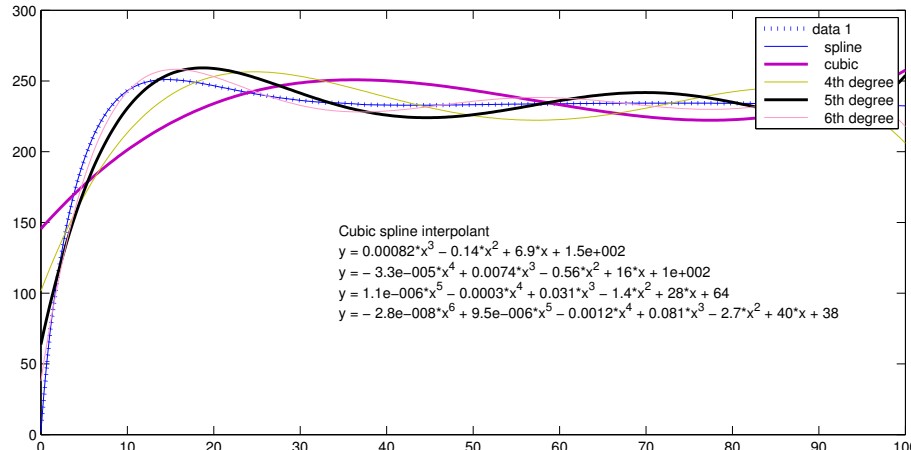

**Figure 1.** Polynomials fitted against the observed data set of rotational curve in galactic halo region (Table 1). The horizontal axis represents distance in kpc and the vertical axis represents the rotational velocity in galactic halo in km/s.

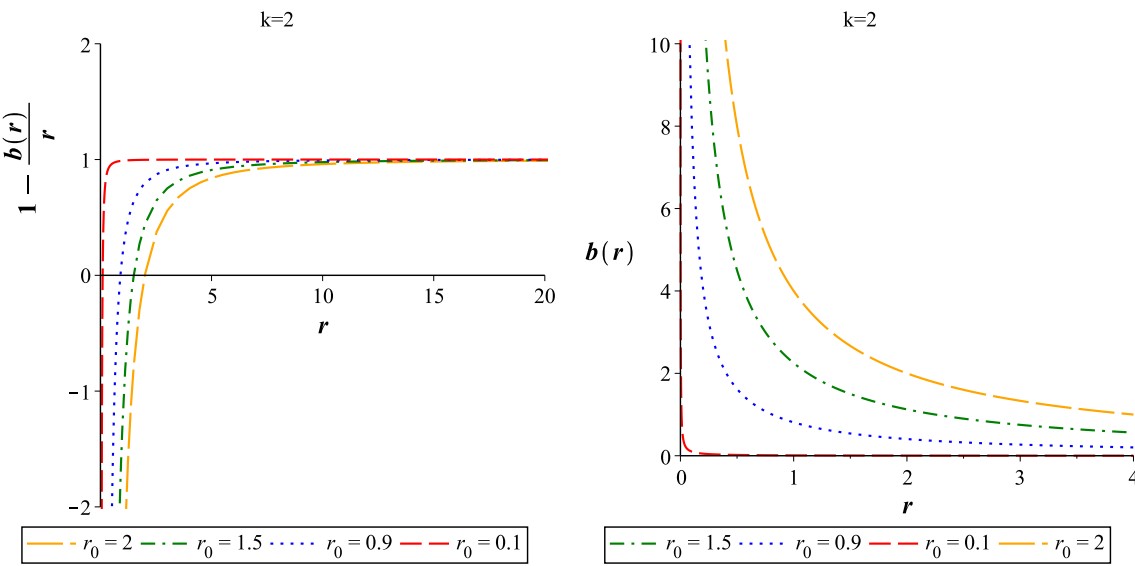

**Figure 2.** Case I: Plot to study $(1 - \frac{b(r)}{r})$ and $b(r)$ for different choices of $r_0$.

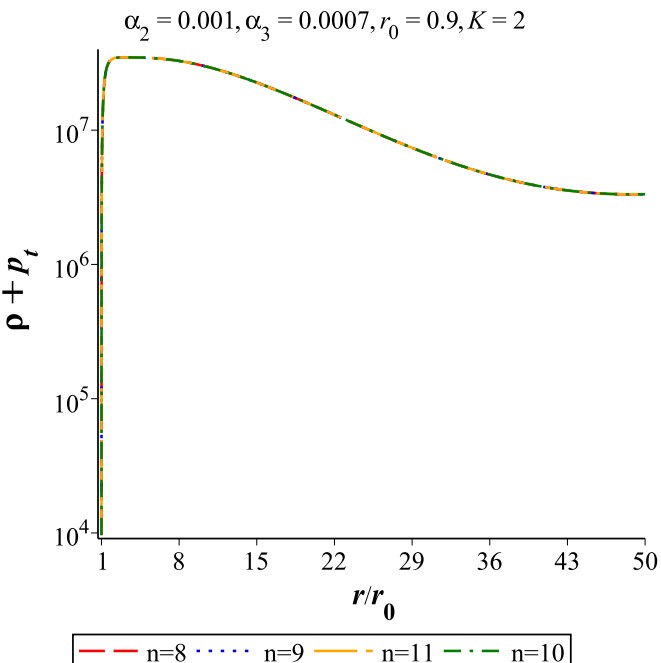

**Figure 3.** Case I: Plot to study $\rho + p_t$ for $n = 8$, $n = 9$, $n = 10$, $n = 11$. It is verified that plots are the same for all sets of values of the constants $\alpha$, $r_0$ and $K$.

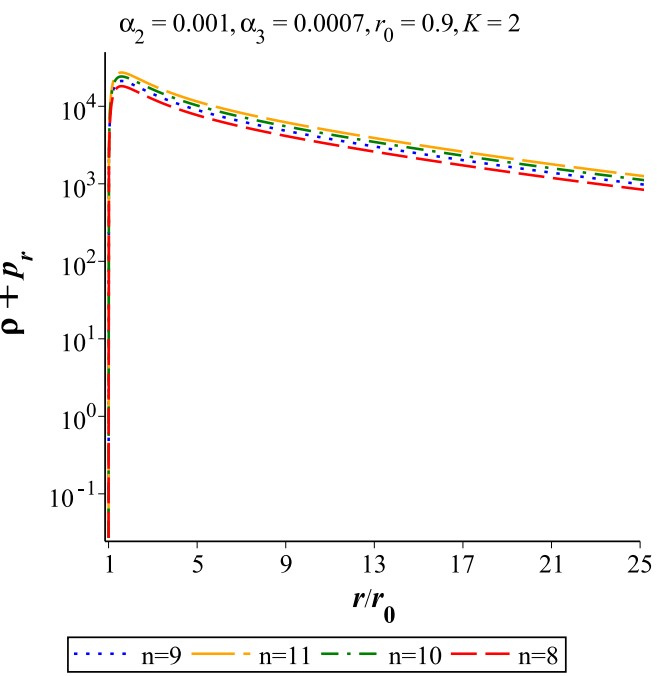

**Figure 4.** Case I: Plot to study $\rho + p_r$ for $n = 8, n = 9, n = 10, n = 11$. It is verified that plots are the same for all sets of values of the constants $\alpha$, $r_0$ and $K$.

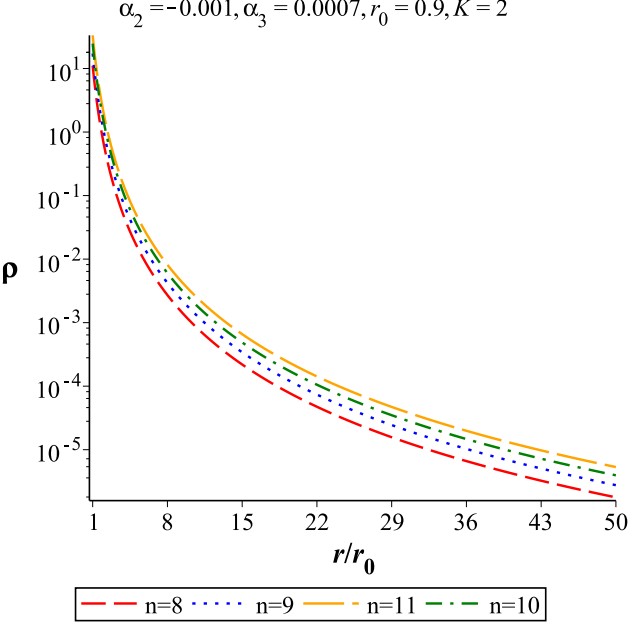

**Figure 5.** Case I: Plot to study $\rho$ for $n = 8, n = 9, n = 10, n = 11$. It is verified that plots are the same for all sets of values of the constants $\alpha$, $r_0$ and $K$.

*4.2. Case II: Physical Features for the Shape Function $b(r) = r_0 \left( \frac{r}{r_0} \right)^k$*

Keeping the previously mentioned conditions in mnd, let us consider another form of the shape function which is as follows:

$$b(r) = r_0 \left( \frac{r}{r_0} \right)^k, \tag{21}$$

where $r_0$ and $k$ are arbitrary parameters as mentioned in the earlier case. Here, the flare out condition of the wormhole implies that $k < 1$.

The present form of the shape function is an increasing function of the radial coordinate $r$ (See Figure 6). In this sense, it is opposite in nature to the shape function in Case I (See Figure 2). Now the physical features that can be noted from Figures 6–9 are as follows:

(1)  The throat of the wormhole may be taken to be at $r = r_0$. The plot of $(1 - \frac{b(r)}{r})$ against the radial coordinate $r$ (Figure 6) clearly shows that $\frac{b(r)}{r} < 1$ for $r > r_0$. Moreover, $b(r_0) = r_0$ where in the plot under consideration $r_0 = 1.72$. These results essentially point out that the flare out condition is satisfied near the throat. It has already been pointed out that there is no horizon in the spacetime.

(2)  The density function (vide Figure 7) again shows a inverse square fall with radial coordinate in the galactic halo region and $\rho > 0$.

(3)  Contrary to the previous case, in the present Case II, the Figures 8 and 9 show that $\rho + p_r \leq 0$ and $\rho + p_t \leq 0$ near the throat indicating the existence of exotic matter that violates the NEC. The plots $\rho + p_t$ remains the same for $n = 8, 9, 10, 11$. Plots are unaffected by different choices of $\alpha_2$ and $\alpha_3$. The plots of $\rho + p_r$ remains similar in nature for $n = 8, 9, 10, 11$ irrespective of the choices of $\alpha_2$ and $\alpha_3$. Thus, we note here a very interesting result that a galactic wormhole in the halo region can exist with normal matter as well as exotic matter under the framework of Lovelock gravity.

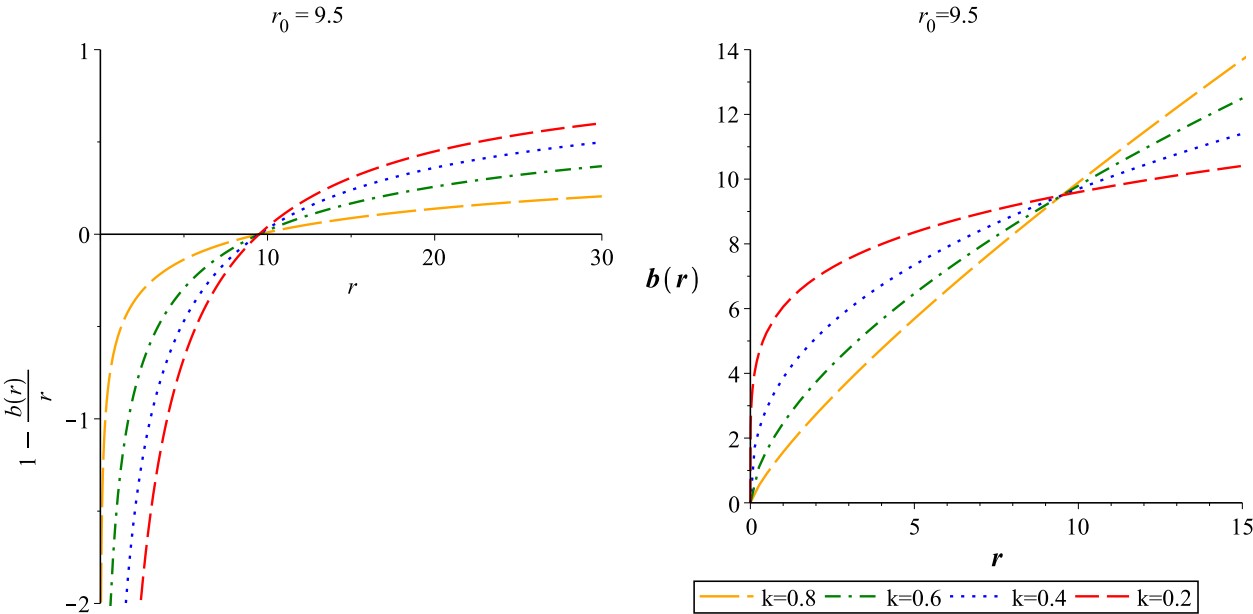

**Figure 6.** Case II: Plot to study $(1 - \frac{b(r)}{r})$ and $b(r)$ for $k = 0.2, 0.4, 0.6, 0.8$.

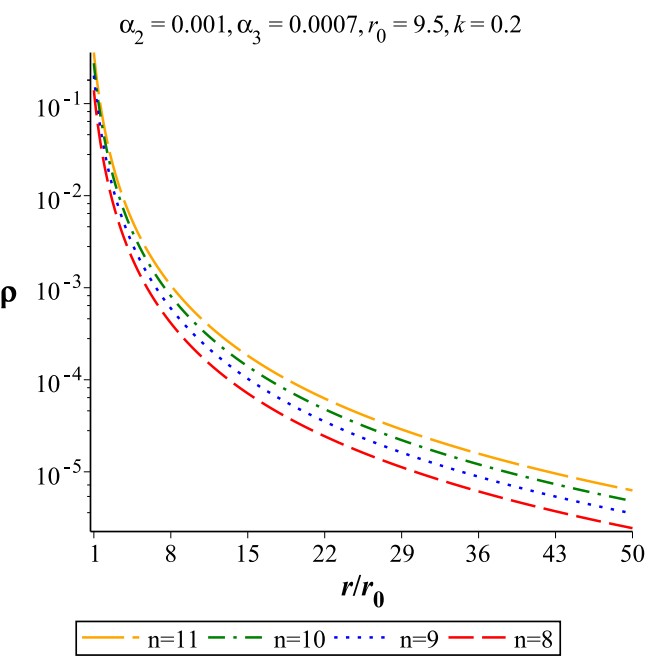

**Figure 7.** Case II: Plot to study $\rho$ for $n = 8$, $n = 9$, $n = 10$, $n = 11$. It is verified that plots are the same for all sets of values of the constants $\alpha$, $r_0$ and $K$.

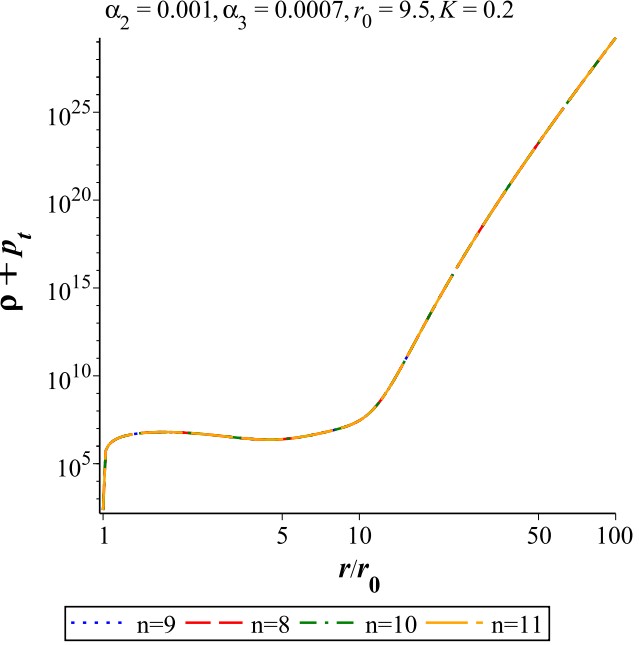

**Figure 8.** Case II: Plot to study $\rho + p_t$ for $n = 8$, $n = 9$, $n = 10$, $n = 11$. It is verified that plots are the same for all sets of values of the constants $\alpha$, $r_0$ and $K$.

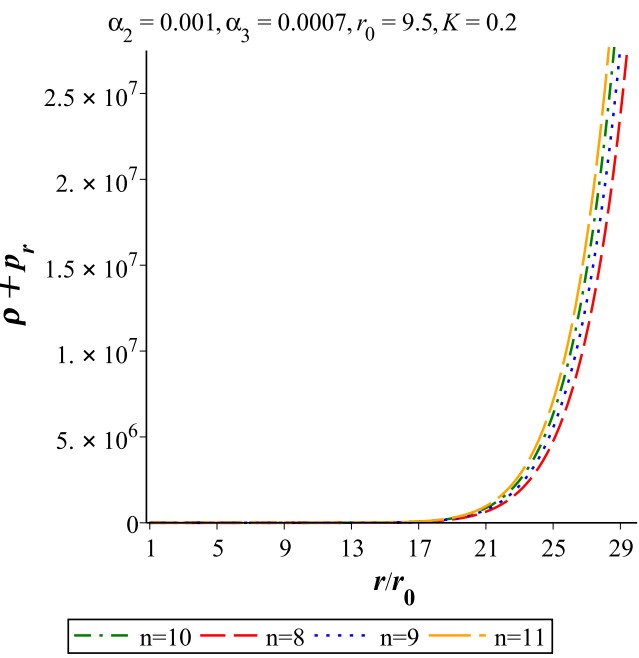

**Figure 9.** Case II: Plot to study $\rho + p_r$ for $n = 8$, $n = 9$, $n = 10$, $n = 11$. It is verified that plots are the same for all sets of values of the constants $\alpha$, $r_0$ and $K$.

## 5. Concluding Remarks

In this paper, our motivation was to extend the works of Rahaman et al. [29,33,35,36] to construct WH model in the galactic halo region on the platform of Lovelock gravity. To do, so we specifically employed the third order Lovelock gravity. By using the cubic spline interpolation technique (see the Appendix A), the rotational velocity of test particles has been found out in the halo region of our galaxy from the observed values of the radial distances and rotational velocities. Considering this value of the rotational velocity, we present possible existence of WH in galactic halo region under the Lovelock gravity.

In favour of this successful execution we are now putting forward some of the salient features based on the results obtained from the proposed model as follows:

(i)   The redshift function calculated from the consideration of the flat rotation curve of the galaxy (Figure 1) remains finite everywhere which clearly implies that there would be no event horizon and thus provide an important criterion for the existence of WH.

(ii)  It is interesting to note that plots on the left panel of Figure 2 (Case I) exhibit presence of the shape function which satisfies the condition $\frac{b(r)}{r} < 1$ for the constraint $r > r_{th}$ and thus indicates that the flare out condition of the WH $b'(r_{th}) < 1$ is fulfilled. For the other form of the shape function (Case II), we observe an increasing function of the radial coordinate $r$ (Figure 6) and hence the feature is opposite in nature to the shape function of Case I (i.e., Figure 2). If we consider the throat of the wormhole to be at $r = r_0$ then the plots of $(b(r) - r)$ against the radial coordinate $r$ demonstrate that $\frac{b(r)}{r} < 1$ for $r > r_0$ which is in confirmation of fulfilling of the flare out condition near the throat.

(iii) Figures 5 and 7 show very highly dense region near the centre of the galaxy and its value very quickly decreases to a small value with increasing distance which is in accordance with the earlier predictions [62–65].

(iv)  Figures 3 and 4 show that the NEC is satisfied in the galactic halo region which is in contradiction to the result under general relativity [29,32,33,35,36]. This may be considered as exception for Lovelock gravity [11,13,17,61]. On the other hand, Figures 8 and 9 show that the NEC is violated in the galactic halo region which is in accordance with the result under general relativity [29,32,33,35,36]. Thus, we note a

very important result in the present investigation that galactic wormhole in the halo region can exist with normal matter as well as exotic matter.

However, the present results intrinsically raise the following awkward issues: (i) is exotic matter really indispensable for constructing traversable WH, and (ii) if not, as suggestions evolve from the modified theories of gravities that galactic dynamics of massive test particles can be explained without introducing any exotic dark energy [66–72], then before coming to a conclusive decision, further studies are needed to perform. These issues may be addressed in a future project under Lovelock gravity as well as other modified gravities in a series of exclusive investigations. This endeavor may shed light on the debate as to whether exotic matter (as GR advocates) is indispensable for constructing a traversable wormhole or alternative gravity theories (especially Lovelock gravity under the present treatment) are suffice to predict WH.

Another point to comment on here is related to evidence of WH in reality, which at present seems belongs to futuristic observational projects only. However, recent detection of black holes [73–76] obviously provides glimpse of hope and aspiration regarding observational signatures for WH also. As a special note, we would like to present here the work of Piotrovich et al. [77] where they have proposed a test for WH, considering an AGN at the centre of a gaclactic core as one mouth of a WH, to perform a comparative study between the $\gamma$ ray spectrum originating from the jets of a supermassive black hole and that from a WH acting as mouth of an AGN. Therefore, under this proposal, they conclude that an observation of such radiation would serve as evidence of the existence of wormholes. A very recent work on the possibility of generalized wormhole formation in the galactic halo due to dark matter using the observational data within the matter coupling gravity formalism also can serve a purpose in this detection issue [78].

**Author Contributions:** Conceptualization, K.C.; methodology, F.R.; formal analysis, S.R.; writing— original draft preparation, B.S.; writing—review and editing, D.D. All authors have read and agreed to the published version of the manuscript.

**Funding:** This research received no external funding.

**Data Availability Statement:** Not applicable.

**Acknowledgments:** K.C., F.R. and S.R. are thankful to the authority of Inter-University Centre for Astronomy and Astrophysics, Pune, India for providing them Visiting Associateship under which a part of this work was carried out.

**Conflicts of Interest:** The authors declare no conflict of interest.

## Appendix A. Cubic Spline Interpolation Method

Cubic Spline Interpolation Method is used to interpose a specific type of piecewise polynomial called spline. Spline interpolation follows the procedure of fitting a curve inside the trend of observational data minimizing the error within a given set of data point. This technique interpolate with minimized error when lower degree polynomial is fitted and the same can be made huge when we use higher degree polynomial. The cubic spline method is constructed with piecewise 3rd order polynomial which have to pass the observational data set called control points avoiding the Runge's phenomenon where oscillation occurs between points. There are advantages of cubic spline interpolation here. Firstly, it improves fitting of a curve by increasing the degree of fitted polynomial. Moreover, with respect to a linear interpolation cubic spline interpolation produces a rather sharp result if the data sets are not well spaced. In the present article, the cubic spline method is constructed based on the 5th degree polynomial, which pass through the observed data set. We used the MATLAB directly to implement the cubic spline interpolation based on the observation. In Figure 1, blue dotted line is for given set of data points for rotational velocity curve ($v^t$) and different splines are arranged to fit against the observed data in the mentioned figure. The fifth degree polynomial is found to be the best fit.

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
