# Peer review of "Galactic Wormhole under Lovelock Gravity"

_universe, doi:10.3390/universe8110581_

Round 1

Reviewer 1 Report

The authors face the fascinating problem of whether wormholes may indeed exist and explain astrophysical data coming from our Galaxy. The paper is very well written (apart from some minor issues that I shall tell about in a moment), the analysis is simple and clear, the results are clearly exposed, and the final result is quite important. Indeed, it seems to suggest that, if wormholes exist, then Lovelock gravity would imply that no exotic matter is necessary.

I definitively recommend publication and wish the authors success in this important line of research. Nonetheless, I would suggest:

i. to pay a bit more attention to little details such as 

- the wrong measure in the integral (1) (it should be d^n x etc, not d^{n+1}x etc)

- the r in the caption of table I should be capital R

- after eq (14) mysterious equations (III) and (III) are recalled whereas I believe they should be (7) and (8) etc

This is important not to convey an idea of sloppiness.

- the figures are not easily readable, they should be made bigger

ii. again on figures, but this time on a slightly more important level, I must ask to make an effort to reduce them! E.g., figure 3 is made of four plots that are identical... why not make one single plot and say that it is the same for all those sets of values of the constants alphas? and the same for figures 4, 5, 7, 8 and the two sets of plots of figure 9

iii. the most important request, though, is to elaborate more on the very important final remarks on the necessity (or not) of the exotic dark energy. The authors show quite an extensive knowledge of the literature on the general area. It would be nice and due, for the readers, to see how this result they have impact on the general discussion. If that means, for instance, that that Lovelock gravity is better equipped than Einstein's, if wormholes are indeed more possible in this framework, etc. Also, what the authors suggest as possible dedicated tests of this, etc.

To summarize, I shall be happy to see the paper published. If the suggestions are followed, I shall be even happier.

Author Response

\textbf{Comment 1(i):} the wrong measure in the integral (1) (it should be $d^n x$ etc, not $d^{n+1}x$ etc)\\

 {Authors response:}} We are very thankful to the respected referee for pointing out the typos which is now rectified. \\

\textbf{Comment 1(ii):}  the r in the caption of table I should be capital R.\\

 {Authors response:}} The indicated change has been done. \\
\\
\textbf{Comment 1(iii):}  after eq (14) mysterious equations (III) and (III) are recalled whereas I believe they should be (7) and (8) etc.\\

 {Authors response:}} Thanks to the respected referee for indicating the typos due to wrong Latex commands which are now rectified. \\

\textbf{Comment 2(i):} the figures are not easily readable, they should be made bigger.\\

 {Authors response:}}  The indicated modifications have been done in the readable size by making them bigger. \\

\textbf{Comment 2(ii):} again on figures, but this time on a slightly more important level, I must ask to make an effort to reduce them! E.g., figure 3 is made of four plots that are identical... why not make one single plot and say that it is the same for all those sets of values of the constants alphas? and the same for figures 4, 5, 7, 8 and the two sets of plots of figure 9.\\

 {Authors response:}} The indicated modifications have been done in the physical reality by making them smaller in numbers. \\

\textbf{Comment 2(iii):} the most important request, though, is to elaborate more on the very important final remarks on the necessity (or not) of the exotic dark energy. The authors show quite an extensive knowledge of the literature on the general area. It would be nice and due, for the readers, to see how this result they have impact on the general discussion. If that means, for instance, that that Lovelock gravity is better equipped than Einstein's, if wormholes are indeed more possible in this framework, etc. Also, what the authors suggest as possible dedicated tests of this, etc.\\

 { {Authors response:}} We have discussed on the above mentioned two issues in the last part of the Concluding section (in BLUE color) as far as possible by providing logic along with citing by several references.

Reviewer 2 Report

Dear Universe editors,

I have read the paper "GALACTIC WORMHOLE UNDER LOVELOCK GRAVITY". 

In this manuscript the authors study a galactic wormhole model within the context of Lovelock gravity. In particular, the authors study, using numerical computations, the physics of wormholes in third order Lovelock gravity.

Interestingly, the authors claim that, in the studied model, horizonless wormholes can exist with normal matter as well as with exotic forms of matter.

I believe that the results presented in this manuscript would be valuable for researches in the fields of gravitation and high-energy physics. I can therefore recommend publication of the manuscript in Universe.

However, before publication the authors should make the following important changes in the manuscript:

 [1] The authors should correct the sentence “Now we can substitute Eqs. (12), (13) and (14) in Eqs. (III), (III), (9)” just after Eq. (14).

In particular, it is not clear what is Eq. (III).

[2] In order to improve the quality of the paper and for the benefit of the readers, the authors should provide a short description of the physical motivation for considering the specific functional forms (15) and (21) for the shape function.

Yours sincerely.

Author Response

Comment [1]: The authors should correct the sentence “Now we can substitute Eqs.
(12), (13) and (14) in Eqs. (III), (III), (9)” just after Eq. (14).
In particular, it is not clear what is Eq. (III)..
Authors response: We are very thankful to the respected referee for pointing out the
typos due to wrong Latex commands which are now rectified.
Comment  [2]:  In order to improve the quality of the paper and for the benefi tof
the readers, the authors should provide a short description of the physical motivation for
considering the specifi cfunctiona lform s(15 )an d(21 )fo rth eshap efunction.
Authors response: We have provided a brief discussion on the above mentioned issue
in the appropriate position of the section (in GREEN color) as far as possible by providing
logic along with citing by several references.